# Consumer Attitudes towards Food Preservation Methods

**DOI:** 10.3390/foods11091349

**Published:** 2022-05-06

**Authors:** Paulina Guzik, Andrzej Szymkowiak, Piotr Kulawik, Marzena Zając

**Affiliations:** 1Department of Animal Products Processing, Faculty of Food Technology, University of Agriculture in Kraków, ul. Balicka 122, 30-149 Kraków, Poland; paulina.guzik@urk.edu.pl (P.G.); kulawik.piotr@gmail.com (P.K.); marzena.zajac@urk.edu.pl (M.Z.); 2Department of Commerce and Marketing, Institute of Marketing, Poznań University of Economics and Business, ul. Niepodległości 10, 61-875 Poznań, Poland

**Keywords:** consumer perception, preserved food acceptance, food preservation, preservation method, novel methods of food preservation

## Abstract

The development and scope of using various food preservation methods depends on the level of consumers’ acceptance. Despite their advantages, in the case of negative attitudes, producers may limit their use if it determines the level of sales. The aim of this study was to evaluate the perception of seven different food processing methods and to identify influencing factors, such as education as well as living area and, at the same time, to consider whether consumers verify this type of information on the labels. Additionally, the study included the possibility of influencing consumer attitudes by using alternative names for preservation methods, on the example of microwave treatment. The results showed that conventional heat treatments were the most preferred preservation methods, whereas preservatives, irradiation, radio waves and microwaves were the least favored, suggesting that consumers dislike methods connected with “waves” to a similar extent as their dislike for preservatives. The control factors proved to significantly modify the evaluation of the methods. The analysis of alternative names for microwave treatment showed that “dielectric heating” was significantly better perceived. These research findings are important as the basis for understanding consumer attitudes. Implications for business and directions of future research are also indicated.

## 1. Introduction

According to European Union law, producers are obliged not only to provide information regarding nutritional value on food and beverage packaging, but also of the specific treatment that has been used on the food product [1]. As consumer awareness increases, so does their attention to what food products they purchase by reading the labels, which often influences purchasing decision [2]. Products which are supposed to be beneficial for people’s health are more likely to be chosen. The most frequently checked consumer information on the packaging is the expiration date, ingredients list and the nutritional value [2,3,4,5,6], whereas the processing method is the least searched [7]. Consumers nowadays seek for products that are fresh, tasty, do not contain preservatives or other chemical additives but, at the same time, are safe and have a long shelf-life. Many people prefer natural foods with “green labels” [8,9,10]. The food producer must adapt to the preferences of consumers and earn their trust, to gain their acceptability [6,11]. The answer to the needs of consumers regarding the high nutritional values and clean label trend is the use of novel technologies in the food industry. The lack of consumer acceptance analysis may determine the high failure rate of novel product implementation, which is why it is crucial to appropriately communicate the use of innovative technologies to consumers [12].

Despite their growing awareness, many consumers exhibit fear of innovative technologies and may show resentment towards novel food products produced with the use of such technologies. This phenomenon is called food technology neophobia and is connected with the cult of natural or traditional food [13,14,15]. Additionally, this is a personality trait that affects consumer willingness to accept new food technologies [16]. There is a belief that traditional, simple food, without preservatives, has the highest nutritional value and is of best quality, which is why such products are highly desirable [11,17]. Moreover, consumers are often deterred by the complicated names of technologies that are unknown and incomprehensible to them [16,18]. Other reasons for the emergence of food technology neophobia are disgust, reluctance towards sensory feature, and fear of danger after consuming this product. Many factors have impact on this phenomenon, including psychological barriers, knowledge, functional barriers connected to ease of use, benefits and risks as well as socio-demographic factors [13]. Furthermore, education level and appropriate dissemination of information are important factors for facilitating the widespread adoption of new food technologies and avoiding failure in innovative marketing strategies [19].

Therefore, production processes and marketing activities should be combined in such a way that consumers perceive it as a product that is as natural and familiar to them as possible [10]. Recent years have shown significant technological progress in the food sector, thereby increasing the quantity of new products on the market [20,21]. Despite the reluctance of consumers to accept new technologies, food products created using such methods may have a positive effect on the product choice. The advantages of using new technologies are: increased production efficiency, increased safety and nutritional value with less demand for energy, water and chemicals, which is also important for sustainability [16,22].

The goal of food preservation is to provide microbiological safety and extend the shelf-life of products. Due to its effectiveness, despite the development of novel technologies, conventional thermal preservation methods are still the most commonly used approaches in the industry [23]. Pasteurisation consists of heating the product up to the temperature of 100 °C, while during sterilisation, the temperature exceeds 100 °C. In conventional thermal preservation methods, heat is generated outside of the material and occurs via convection throughout its entire volume [24]. Despite guaranteeing microbiological safety, these traditional methods result in reduction of some thermally sensitive food ingredients, especially vitamins and polyphenols, which are related to food quality [25].

Microwave heating is a valid alternative to conventional heating. Microwaves are radio waves in the spectrum of electromagnetic radiation with wavelengths ranging from one millimetre to even one metre. In literature on the subject, microwave heating is often referred to as dielectric heating because waves are absorbed by materials having dielectric properties [26,27]. Such materials, also called dielectrics, have a relatively high specific resistance and low electrical conductivity. Moreover, the molecules or atoms comprising the dielectric (such as agri-food products) exhibit dipole movement. Microwave technology is a widely used technology within the food industry. It is applied for cooking, drying, thawing, pasteurisation or sterilisation of food products and most households contain a small microwave [28,29,30,31].

There are also non-thermal technologies that extend the shelf-life of food products without the use of high temperatures. One of such is the use natural bioactive compounds such as using plant materials. Some plant extracts show antioxidant and/or antimicrobial properties and can be used as an alternative to chemical preservatives [32]. Another example are chitosan-based apple peel polyphenols [33] furcellaran-chitosan [34] or furcellaran-gelatin with green and pu-erh tea edible coatings [35]. Physical non-thermal preservation methods allow reduce treatment time.

Different non-thermal preservation technologies include ionising radiation, which is also in the range of electromagnetic waves [25]. Irradiation of food uses low-energy radiation and 3 types of radiation are authorised for use in the food industry, which include high-energy gamma rays, X-rays and accelerated electrons in accordance with the Codex General standard for Irradiated Food [25,36]. Irradiation works directly through damage of cell components such as carbohydrates, DNA and lipids, and indirectly via free radicals and reactive species (e.g., hydro-electrons, hydrogen atoms, or hydroxyl radicals). It is a consequence of radiolysis of water reaction with cells or food components, and has proven effective in the reduction of insects and microorganisms [37,38].

High hydrostatic pressure (HHP) is a technique in which pressures of 100–1000 MPa are applied to the product, resulting in the inactivation of the majority of pathogenic microorganisms. HPP-treated food must be packed and contain water, then closed in a chamber where the pressure is gradually increased. It is closed in a chamber where the pressure is increased. Depending on the type of food, the process lasts from several seconds to 20 min [39,40]. Consumer demand for freshness and the longest possible shelf-life can be provided by packaging in modified atmospheres (MAP). This technology is based on packaging in vapour- or gas-barrier materials of fresh and minimally processed food products in a packaging system where the air composition is changed, ensuring an optimal gas composition around the product. Due to reduced O_2_ and CO_2_, the metabolic processes and microbial activity significantly slow down, thus extending the shelf-life of the food product [41,42].

The abovementioned techniques are only several examples of the modern methods already in use within the food industry and producers apply them to preserve their products. The applicability of the methods may depend on many factors and also have their pros and cons. The advantages and disadvantages of the described technologies are presented in Table 1. Despite growing awareness, many consumers do not read labels for various reasons, such as lack of time, too much information on the label or trust in the brand name [5,43]. From the producer’s point of view, consumers who declare reading labels are credible in assessing the product, and it can also be assumed that they understand the declarations on the label better. The results of the study carried out by Szymkowiak et al. [7] allowed to show a specific attitude-behaviour dissonance in which consumers declared that the most important attribute of the product choice is the processing method, even though it was the least searched for on the label.

Despite the many advantages and electromagnetic heating being widespread, especially the microwave spectrum, some consumers still negatively perceive this kind of preservation. Negative associations are related to radiation and it being harmful for human health [3,57,58,59]. One of the potential ways to improve microwave perception among consumers is changing the method name to another “safe sounding” synonym. Associations of the preservation method with a product influence purchasing decision [60,61]. Therefore, the aim of this research study was to evaluate the perception of different food processing methods and to identify factors influencing them. In the study, the possibility of influencing the attitudes of consumers by modifying the processing name was also included.

## 2. Materials and Methods

### 2.1. Respondents

In order to assess the preferences of various food preservation methods, a questionnaire study was conducted among 438 respondents who declared that they were responsible for making food purchases in their households. The survey was conducted both in electronic and paper version to reach respondents from rural areas, who do not actively use the Internet. In both cases, the participants did not receive any remuneration. Participants who failed the validation questions (e.g., “If you read the question carefully, select the answer I strongly disagree”, *n* = 33) were excluded from the analyses. Therefore, the answers of 405 respondents were taken for analysis. The mean age in the analysed sample was 34.71 (SD = 1.85, minimum = 19, maximum = 85). The majority were women (74%). The group differed in terms of household and place of residence size. The detailed characteristics of the sample are presented in Table 2.

### 2.2. Study Design

The questionnaire was divided into 3 parts. In the first one, the respondents were to indicate their attitude towards the analysed methods (Table 3) on a 7-grade scale where 1—means a very negative attitude and 7—a very positive attitude. In the second part of the study, a photo of the product (ham) was presented 4 times together with an annotation that the product was preserved by 1 of 4 methods: electromagnetic wave with a length of 32.76 cm (wavelength of microwave at 915 MHz frequency), dielectric wave, electromagnetic wave or microwave. These terms are synonymous with each other from a technological perspective; however, linguistic modifications were assumed to be associated with different reactions and thus, different perception of the product. The order of displaying the methods was random, which allowed to eliminate bias. Participants were not provided with additional definitions or explanations of the methods. This allowed their overall relationship with the methods to be assessed, which is consistent with the level of detail shown on product labels. On this basis, consumers were to state how much they would be interested in purchasing a product on a seven-point scale (1—“definitely not”, 7—“definitely yes”). Finally, the respondents answered the identification questions, including those regarding their purchasing behaviour.

### 2.3. Data Analysis

At the stage of data analysis, ANOVA with repeated measurements was used. This type of analysis of variance, to a greater extent, allows to take the variability for a particular respondent into account. Thus, it is possible to identify differences, also in a situation where some respondents generally expressed higher preferences for all methods, or vice versa. Subsequently, additional analyses were conducted between groups due to additional moderators important from the perspective of the subject under study: education and declared paying attention to processing methods when selecting food products at a store. In the case of education, the division was made into 3 groups: people with secondary and lower education (*n* = 129), bachelor level (138), and master’s degree level and higher (138). The χ^2^ analysis showed that the differences in group sizes were statistically insignificant: chi^2^ (2) = 0.04, *p* = 0.98. In the case of the second variable, two groups were created: people who verified (155 respondents) and did not verify the type of preservation method (250).

## 3. Results and Discussion

### 3.1. Consumer Preference for the Food Product Preservation Method

In the first part, ANOVA analysis was performed in 3 iterations. The first analysis allowed to confirm that the method strongly determined preferences F (2828, 7) = 271.159, *p* < 0.001, η^2^ = 0.402. Post-hoc analysis, applied to compare the obtained values for all pairs, showed significant statistical differences between them, apart from the relationship between MWV and RWV (t(404) = 2.192, *p* = 0.057) and between in RWV and PRE (t(404) = 1.702, *p* = 0.089). Detailed values for all comparisons are presented in tabular form in Appendix A. The most negative ratio (average below 3 on the 1–7 scale) was indicated by the respondents in relation to IRR (M = 1.975, SD = 1.379), PRE (M = 2.235, SD = 1.348), RWV (M = 2.398, SD = 1.462) and MWV (M = 2.607, SD = 1.501). The respondents expressed a more positive attitude towards MAP (M = 3.672, SD = 1.868), HPP (M = 4.052, SD = 1.790) and STE (M = 4.306, SD = 1.763). The PAS method was rated the highest (M = 4.975, SD = 1.613). Next, an analysis was conducted with the use of a moderator, which was the declared behaviour in the store. A table with results of post-hoc analysis, depending on in the store behaviour, is provided in Appendix A. This analysis revealed that the factor significantly influences the general level of preferences in relation to all methods (F (403.1) = 5.423, *p* = 0.02, η^2^ = 0.003). People who check food preservation methods on products when shopping at a store express a more positive attitude towards all of the methods (Figure 1).

The results of the study indicate differences between various methods of food preservation and consumer preferences. Consumers’ nutritional literacy affects their ability to process food labels [62]. Therefore, it can be assumed that consumers who declared verifying the preservation method on the label of food products have a better understanding of the method itself. In our study, consumers who verified the preservation method gave higher ratings for preservation method preferences, but the trend was the same for non-verifying consumers.

In this study, PAS and STE were conventional types of preservation methods and both of them received the highest score. Consumers are accustomed to the above-mentioned techniques because they associate them with traditional production to which they are accustomed [63,64]. Moreover, they were programmed from early childhood to prefer familiar foods [65]. Conventionally processed products are mainly associated with health and natural products, while those including references to industrial processing technology are perceived as processed and, therefore, unhealthy [66,67]. MAP and HHP were also rated relatively high, which may mean a positive reception of these technologies. In previous research conducted by Deliza and Ares [12], it was also confirmed that consumers have a positive attitude towards HHP and are willing to buy products treated via this technology, after being informed about how this method works. Nonetheless, it should be noted that many people have never heard of this technology. In a study by Lee et al. [68], the respondents evaluated juices treated with HHP and pulsed electric fields. In their opinion, conventional thermal treatment decreased the pleasant notes such as natural and fresh, whereas undesirable notes like cooked flavour or sourness increased. Similar to the case of MAP, Guerrero [11] demonstrated that this technology was negatively perceived by consumers. Participants were suspicious and immediately rejected the MAP packed product. Moreover, it was noted that they are not willing to pay more for products packed in MAP, despite the fact that those products maintained freshness and high quality for a longer period of time. Ortiz et al. [69] indicated that consumers can pay more for vacuum packaging as opposed to MAP.

The significance of naturalness has crucial meaning for consumers nowadays. They prefer food free from preservatives, additives or artificial ingredients for perceived naturalness of foods. The result is that now, more than ever, manufacturers often try to produce products with “green labels” [10]. A perceived lack of naturalness also hinders the acceptance of new food preservation methods and technologies [65]. This phenomenon would explain the negative perception of preservatives in our research. In a study by Perito et al. [67], the majority of the respondents declared willingness to consume biopreservatives, only if they replaced synthetic ones. According to Dominick et al. [70], 83% of respondents agreed that a product with an “all natural” label meant no preservatives. They perceived “all natural” foods without preservatives and additives as products with better taste, nutritional value and increased food safety. IRR, RWV and MWV were similarly perceived, relatively negative, as the preservatives. These, technologies have a common denominator, since all 3 methods are based on “waves”. In a survey by Szymkowiak et al. [7], respondents showed dislike towards the microwave preservation method, whereas conventional thermal preservation was considered the most positive. Microwave technology is known to consumers through the widespread domestic use of microwave ovens. However, there is a common belief among such consumers that microwaved foods are unhealthy and often associated with radiation [58]. The negative perception of microwaves can be a result of disinformation and fake news in social media, such as “using microwaves to heat food can cause carcinogenesis” [71,72]. In their study, Wolfson et al. [73] noted that consumers perceived microwave heating negatively due to radiation or “*zapping*” nutrients out of the food. Especially among the older generation of consumers, their statement was also that microwave heating is “lazy” or “cheating” one’s way out of cooking. Consumer aversion to radiation-related technologies of food preservation may be associated with the risk of making food radioactive or the formation of harmful compounds, but also with the wrong image that food irradiation is a nuclear technology. Moreover, for some consumers, products labelled as irradiated may be read as a health warning [12,61,74,75]. The lack of proper knowledge affects consumer acceptance of food irradiation technologies and can be a main reason for the limited application this method [74].

### 3.2. Consumer Preference for Food Product Preservation Methods According to Education Level and Main Area of Residence

For the education variable, the study revealed between- (F (402.2) = 4.090, *p* = 0.017, η^2^ = 0.005) and within-subject effects (F (2814.14) = 4.122, *p* = 0.001, η^2^ = 0.009). This indicates that the level of education determined the overall level of method preference as well as the interaction effect. People with lower educational levels, compared to other groups, showed lower preferences for most methods, except pasteurisation (Figure 2). A table with the post-hoc analysis values for all 276 pairs of comparisons can be found in Appendix A. Moreover, we observed that the main place of the respondents’ residence determines the level of acceptance of various methods (F (3.401) = 4.874, *p* = 0.002). Residents of rural areas, on average, assessed the methods of food preservation by 0.4 lower on a 7-level scale than residents of cities with 100,000–500,000 inhabitants ((t(404) = 2.662, *p* = 0.04), and compared to residents of cities with above 500,000 inhabitants ((t(404) = 3.444, *p* = 0.004). This means that people living in rural areas have a lower acceptance of various food preservation methods.

The results of our research allowed to show that respondents with the lowest (1 in Figure 2) education level caused polarisation between the options for declaring preferences for preserving methods. It can be interpreted that they definitely prefer the familiar, conventional technologies, and most definitely, do not prefer those which are unknown to them. The group of consumers with the highest education level (3 on the graph) perceived novel methods better compared to those least educated. This may result from the fact that more educated people show greater knowledge and awareness related to novel methods of food preservation. Popek and Halagarda [76] also confirmed correlations between education level, place of residence and greater knowledge of consumers. Similarly, Moreb et al. [77] indicated that people living in the city were more knowledgeable about food safety and food handling practices than those who lived in the countryside.

The survey, in which consumers were asked about their knowledge of microwave radiation and its effect on food, revealed that consumers know very little about it. The reason may be that it is difficult to obtain such knowledge from reliable sources [3,78]. Most consumers who do not know the process or who have little knowledge of it, show greater uncertainty regarding the safety of processed food products and often believe that they are dangerous and may pose a health risk [74]. Verbeke et al. [79] demonstrated that providing additional information about novel technologies positively increased their perception. Nonetheless, preservatives, although known to most people, are nor accepted. Increasingly, manufacturers are resigning from their addition, despite the fact that they are mostly considered safe. However, due to the fact of their potentially negative effects and low level of acceptance among consumers, other solutions or natural substitutes are being sought [80,81,82].

### 3.3. Consumer Preference for Alternative Names of Microwave Treatment

In the last part of the analysis, in accordance with the second goal adopted in the paper, the impact of the alternative name of the method for microwaves on product perception, was analysed. The study revealed that different names cause different perceptions of the product (F (1212.3) = 17.874, *p* < 0.001, η^2^ = 0.042). *Post-hoc* analysis revealed that while interchangeably using MWV, EMV, LWV does not cause statistically significant differences, the DIE version of labelling (M = 2.986, SD = 1.510) results in more desirable reactions (Table 4). Additional analyses carried out with controlling the inter-group factor revealed the importance of both the interest in the processing methods expressed by the declaration of method verification while shopping ((F (403.1) = 8.533, *p* < 0.004, η^2^ = 0.015) and education ((F (402.2) = 4.571, *p* < 0.011, η^2^ = 0.016) on the use of alternative names for general product preferences. Detailed analysis confirmed that consumers, verifying methods while shopping, better perceived the product described as DIE (M = 3.316, SD = 1.445), and this was a statistically significant difference (t(404) = 3.769, *p* = 0.004) compared to the preferences for this version of the product declared by persons who did not verify the method (M = 2.752, SD = 1.511). In the case of education, the mean value of the group with the lowest education was lower (M = 2.767, SD = 1.598) than in the group with undergraduate (M = 3.116, SD = 1.324) and graduate education (M = 3.007, SD = 1.587).

Microwaves are non-ionising electromagnetic waves having a frequency within the range of 30–300 MHz, and wavelength ranging from 1 m to 1 mm. Microwaves are recognised as radio waves and are absorbed at the molecular level. They react with dipoles and ions and have the ability to heat a material with dielectric properties. This explains why various names of the microwave technology (such as “electromagnetic radiation” or “radio-frequency waves preservation”) could be used. The thermal effect of microwaves is obtained through the molecular movement of dipoles and ions, which generates friction among the rotating molecules and, subsequently, in the dissipation of the energy as heat [29,83,84]. Due to the negative perception of microwave treatment [7], in this study, 4 alternative names were compared (microwave preservation—MWV; preservation using am electromagnetic wave with a length of 32.76 cm—LWV; electromagnetic wave preservation—EMV; and dielectric heating preservation—DIE). MWV, LWV and EMV showed no significant differences in consumer perception. Only dielectric heating preservation was significantly better perceived among alternative names. This might be because only in this name was there no reference to “waves”. In addition, the prefix “di” refers to multiplication, which can be seen as an enhancing element increasing and improving the attributes of the product. Moreover, consumers might associate them more with the traditional methods of meal preparation using an electric stove or magnetic induction (dielectric and electric). This, however, is just a hypothesis which should be confirmed in future research. Familiar associations of technology name with technologies that can be applied at individual households increase acceptance [18], whereas the term “radiation” in microwave radiation can raise concerns [85]. It should be also mentioned that although dielectric heating preservation was perceived much better than alternative names of microwave treatment, the preference value was still 3.316, which is relatively low compared to the other methods under study, as it was below the lower half of the rating scale (Appendix A). Such a different attitude can also be explained on the basis of feelings-as-information theory [86,87,88]. The ease of processing individual names influences judgment [89]. Subjective experiences such as emotions and metacognitive feelings serve as a source of information for consumers and influence decisions made [87,90,91].

In this study, the importance is shown of terminology used in food technology in relation to consumer perception. Names that evoke positive associations are more preferred by consumers [60,61]. Even without knowing the details of the new technologies, the name itself may affect the perception by consumers and their willingness to buy. On an example of cultured meat, Verbeke et al. [79] explained that possible cognitive association or attitude activation matter, such as “in vitro” may activate attitudes linked with laboratory practices or growth processes in bioreactors. “Technical” names may evoke thoughts or strengthen perceptions of unnaturalness, being too scientific for the consumers. In a study by Martins et al. [66], respondents positively associated the cold-pressed juice concept, although the processing method was unknown to them. They had associations with a natural and unprocessed product, probably due to the words “pressed” and “cold” in the name of the technology.

There are various advantages and benefits that many novel food processing technologies provide for food processors and consumers alike. Despite this, however, many consumers negatively regard these novel food technologies. Changing the name of the preservation method, especially to one which is not associated with “waves”, seems to be a valid alternative for food processors. Additionally, the research results indicate that for producers, the application of the methods may also be conditioned by the target group and market. This can increase consumer trust and willingness to buy, while maintaining safety and often higher nutritional value of the product.

## 4. Conclusions

The implementation of new technologies is an indispensable element of food engineering and the production of new food products. Novel food processing and preservation technologies offer many advantages in terms of nutrient retention, good quality and food safety. However, the implementation of new technologies is strongly related to acceptance by consumers. The results of this study allowed to indicate that conventional methods of food preservation were best perceived and accepted by respondents as they are well-known to them. Packaging in a modified atmosphere and with high hydrostatic pressure were also relatively well-perceived, whereas every method related with radiation (microwaves, radiowaves, irradiation) were perceived negatively, comparably to preservatives. Higher-educated respondents perceived new technologies more positively, which may result from their greater awareness and knowledge of food preservation. In the evaluation of the preferences for alternative methods to microwave treatment, the respondents rated “dielectric heating” as the best compared to the alternative “waves” methods. Consumers, despite their interest in new methods of food processing and preservation, are still distrustful of new technologies and worry about food safety. Increasing efforts regarding education about these technologies should result in their acceptance. Moreover, future research could be conducted to investigate, what, apart from the name of the preservation technology, would convince the consumers to increase their preferences for novel food preservation technologies. The authors, despite their best efforts, have not managed to eliminate all the factors that limit the generalization of the obtained conclusions. The study was done both online and on paper, which made it possible to reach people who do not use the Inter-net less or at all, in addition, the data was also collected in rural areas, however, it does not allow to indicate that the sample is representative. The study focuses on the general assessment of individual methods and the identification of indirect factors, such as education or the main area of residence, without referring to the immediate reasons that may determine individual preferences. Understanding the motives is an important next step in understanding consumer behaviour and how to form their attitudes.

## Figures and Tables

**Figure 1 foods-11-01349-f001:**
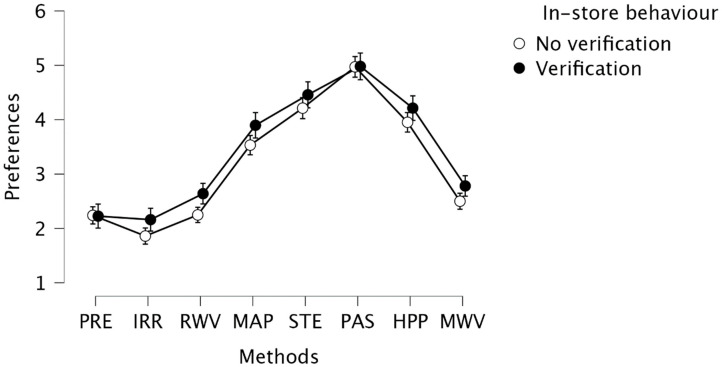
Consumer’ preference for food product preservation methods (PRE—addition of preservatives; IRR—irradiation preservation; RWV—radio wave preservation; MAP—packaging in modified atmosphere preservation; STE—sterilisation; PAS—pasteurisation; HPP—high pressure processing; MWV—microwave preservation).

**Figure 2 foods-11-01349-f002:**
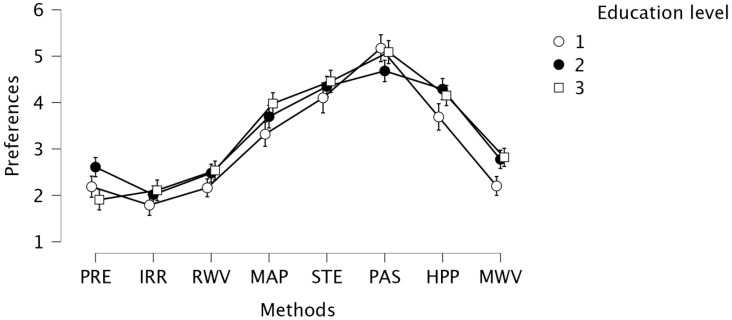
Consumer preference for food product preservation methods depending on education level; 1—lowest; 2—average; 3—highest (PRE—addition of preservatives; IRR—irradiation preservation; RWV—radio wave preservation; MAP—packaging in modified atmosphere preservation; STE—sterilisation; PAS—pasteurisation; HPP—high-pressure processing; MWV—microwave preservation).

**Table 1 foods-11-01349-t001:** Advantages and disadvantages of selected food preservation methods.

Treatment	Advantages	Disadvantages	Source
Conventional pasteurisation and sterilisation	- High effectiveness of preservation.	- Slow heat transfer and long sterilisation time; - Deterioration of colour, texture and flavour; - Significant losses of nutritional value; - High degree of sewage generation; - High cost of application.	[24,44,45]
High hydrostatic pressure	- Non-thermal technology; - Inactivation of vegetative forms of most microorganisms, such as *Salmonella* spp., *Listeria monocytogenes*, yeast, moulds and inactivation of enzymes that cause food spoilage; - Preservation of nutritional value and quality of the product.	- High investment costs; - Inactivation rate can be insufficient, depending on the type of organism and treatment parameters; - Low throughput; - Food containers must be resistant to deformation.	[46,47,48]
Modified atmosphere packaging	- Preserving the stability of fresh or minimally-processed food; - Non-thermal technology; - Can be used in combination with almost any other preservation technique; - Inhibits the growth of microorganisms as well as oxidation progression; - Prevents discolouration of some products (if appropriate gas mixture is used).	- Increases the packaging cost; - Requires more space during storage; - Packages can be easily damaged resulting in a food safety hazard; - The most favourable gas mixture must be chosen for each product type; - Limited effectiveness, not comparable to conventional pasteurisation and sterilisation.	[41,49,50]
Microwaves	- Operational safety; - Minimal loss of heat-labile nutrients (vitamins, antioxidants, phenols and carotenoids); - Reduced processing time; - Lower energy and water demand; - Volumetric heating.	- Hot and cold spots; - The treated products have to be in regular shapes and of homogenous structure.	[29,51,52,53]
Irradiation	- Microorganism inactivation; - Non-thermal method; - Easy to control; - Can save energy consumption up to 70% to 90%.	- Expensive equipment; - Taste of irradiation when - operating improperly; - Necessity to provide information about using this method on the label in many countries; - Multiple legislator restrictions in many different countries.	[54,55,56]

**Table 2 foods-11-01349-t002:** Description of the study group.

Gender	
Women	301
Men	104
Total	405
Main area of living	
City with up to 100,000 inhabitants	88
City with 100,000–500,000 inhabitants	54
City with above 500,000 inhabitants	128
Rural	135
Total	405
Number of people in the household	
1	41
2	85
3	92
4	96
5	58
above 5	33
Total	405

**Table 3 foods-11-01349-t003:** Designation of the analysed methods.

Food preservation methods	Abbreviations used in the text
Radio Wave Preservation	RWV
Irradiation preservation	IRR
Addition of preservatives	PRE
Packaging in modified atmosphere preservation	MAP
Preservation with high temperatures—pasteurisation	PAS
Preservation with high temperatures—sterilisation	STE
Dielectric heating preservation	DIE
Microwave preservation	MWV
High-pressure processing	HPP
Microwave preservation	MWV
Preservation with electromagnetic wave (32.76 cm length)	LWV
Electromagnetic wave preservation	EMV
Dielectric heating preservation	DIE

**Table 4 foods-11-01349-t004:** *Post-hoc* Comparisons—Alternative names for microwaves (PRE—addition of preservatives; IRR—irradiation preservation; RWV—radio wave preservation; MAP—packaging in modified atmosphere preservation; STE—sterilisation; PAS—pasteurisation; HPP—high-pressure processing; MWV—microwave preservation).

		Mean Difference	SE	*t*	Cohen’s d	Holm’s *p*
DIE	EMV	0.402	0.066	6.121	0.304	<0.001
	LWV	0.356	0.065	5.504	0.273	<0.001
	MWV	0.360	0.069	5.190	0.258	<0.001
EMV	LWV	−0.047	0.050	−0.946	−0.047	1.000
	MWV	−0.042	0.062	−0.678	−0.034	1.000
LWV	MWV	0.005	0.064	0.078	0.004	1.000

## Data Availability

Data is contained within the article or Appendix A.

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
