# Peer review of "Consumer Attitudes towards Food Preservation Methods"

_foods, 2022, doi:10.3390/foods11091349_

Round 1

Reviewer 1 Report

I would recommend to add more sources from WoS and Scopus Databases, add the limits and barriers of the research, provide deeper discussion and once more check the formal side of the paper.

Source of each Table and Figure needs to be added.

Reviewer 2 Report

The purpose of the article was to assess the perception of 7 different methods of food processing and to identify influencing factors such as education as well as living space and at the same time to consider whether consumers check this type of information on labels.
The authors approach the theoretical part in a complete way and present a clear image on the approached subject.
The only significant observation on the text would be related to the size of the sample, namely I would recommend the inclusion of arguments for which the number of 438 respondents was considered representative (relative to what / what is the titular population / statistical population, probability of guaranteeing results, etc.) .
Regarding the appearance of the article and the arrangement on the page, I would suggest the correction from the bibliography (which is numbered twice). In addition, from line 207 to the end, the font is different from the rest of the text (I suggest reformatting).

Reviewer 3 Report

Article :

Title: Consumer Attitudes Towards Food Preservation Methods

Journal: Foods.

Comments to authors:

In general, a sentence shouldn’t be more than three lines and paragraph must be justified

Line number 38 rephrase from start

Line number 40 split the sentence

Line number 44. Please add some more references about innovative nonthermal technologies. Please check these references. 1. Journal of Food Processing and Preservation, 45(1), e15018. 2. Food Science and Nutrition, 9(6), 3048-3058. These papers can also be used in the coming paragraphs.

Line number 55 remove full stop

Line number 73 add reference (Comprehensive Reviews in Food Science and Food Safety, 17(2), 437-457)

Line number 83 add reference (International Journal of Food Science and Technology, 54(8), 2563-2569 & Foods, 9(2), 214.)

Line number 91. add reference

Line number 94-97 split the sentence

Line number 101 incomplete sentence

Line number 102 pressured is gradually increased

Line number 113. add reference

In table, Why treatment is italic, please also start the sentence (should be capital)

Line number 219 add more details in last. It is incomplete sentence

Line number 223 citation way is different from remaining article

Line number 242 replace word level with another word. As statement is confusing

Line number 267 rephrase line (found/observed/noticed)

Line number 344 replace “with” some other word as it used two times in a sentence

Line number 365 add reference

Overall comments

Scientific names should be check again.
